# Professional Ethos and Resilience in the Identity of Practicum Students during the Pandemic Context

Daniel Caballero-Julia [1,*,†], Antonio Sanchez-Martin [1], María José Hernández-Serrano [1] and Lucia Herrarte-Prieto [2]

1   Escuela Universitaria de Magisterio de Zamora, Universidad de Salamanca, 49029 Zamora, Spain; antoniosanchez@usal.es (A.S.-M.); mjhs@usal.es (M.J.H.-S.)
2   Facultad de Educación, Universidad de Salamanca, 37008 Salamanca, Spain; luciahp@usal.es
*   Correspondence: dcaballero@usal.es
†   Current address: Escuela Universitaria de Magisterio de Zamora, Avenida Principe de Asturias s/n, 49029 Zamora, Spain.

**Abstract:** Professional ethos encompasses individual factors such as attitudes and expectations, and commitment and responsibility, as well as other social factors related to collaboration or identification with role models. All of these factors are expressed in professional experiences and routines. Practicum experiences within teacher education programmes contribute to the initial expression and formation of students' professional ethos. This study aims to analyse how the restrictions in the context of the COVID-19 pandemic affected the formation of future teachers' professional ethos and their resilience skills. The study analyses the assessment discourse of the practicum during two academic years affected by the pandemic, involving a sample of $n = 725$ students. The Multivariate Analysis of Variance (MANOVA) Biplot and Reinert's Alceste methods were employed to examine differences in the discourse based on gender and developmental changes over the two academic years. The results highlight the significant impact of limitations on routines as a determining factor. Additionally, both collaboration with professional tutors and innovation stood out as variables in the development of professional ethos. Gender differences revealed more negative expectations among female students, while, per year, there was a more positive evolution in learning expectations among male students. In conclusion, the practicum experiences during the pandemic-affected academic years facilitated the development of students' professional ethos, emphasising resilient attitudes and solutions.

**Keywords:** teacher education; internship; higher education; professional ethos; resilience; COVID-19; MANOVA Biplot

## 1. Introduction

The professional identity of educators begins to take shape during the practical application processes offered in university degree programmes [1]. As the initial professional experience in teacher training, the practicum period facilitates formative learning experiences that contribute to this identity. These experiences are supervised by academic and professional tutors, constituting a dynamic and ever-changing process of self- and co-construction within various teaching contexts. At its inception, this construction process is rooted in prior expectations, experiences, and routines [2].

Practicum periods within teacher education programmes play a pivotal role in the preparation of educators [3]. They enable students to develop their professional skills through observation, active participation, feedback, teamwork, and participation in educational projects. Furthermore, being supervised processes, they provide an invaluable opportunity for students to reflect on the practical experience and deconstruct pedagogical gains [4]. In terms of their role in preparing students for the teaching profession, these experiences allow them to acquire the skills, knowledge, and attitudes necessary to carry out their teaching duties ethically and responsibly.

Some scholars consider practicum experiences during university education as a form of professional socialisation [5]. Vanhulle et al. [6] align professionalisation in higher education with a form of dual linkage movement, giving practical meaning to the disciplinary logic of theoretical subjects while optimising expectations for professional performance. Similarly, Zaouani-Denoux [7] regards the value of practicums as an exceptional integration and professionalisation strategy, enabling students to enter into a dynamic self-construction process.

Returning to Bourdieu [8–11], as a result of the socialization process, what the author called *habitus* is generated. This concept implies a system of structured and structuring dispositions capable of generating a certain way of acting in certain situations. The *habitus* constituted within a social space made up of the sum and nature of cultural and economic capitals, would imply an *ethos* (ethical part) and a *hexis* (physical part). In the same way, it should be remembered that the *habitus* (and consequently the ethos and hexis that compose it) can be understood in a plural manner, among them being, for example, the professional *habitus* [12]. As with any *habitus*, it "is characterized by a process of gradual incorporation into the professional environment through intersubjective relationships" [13]. These relationships are mediated by social variables such as social class or gender, as well as within a specific historical context [14,15].

As Romero and Yurén ([16]) remind us, the concept of professional ethos implies a whole system of dispositions that is constituted and configured in order to address the problems and situations that arise in the context of the daily professional life of a given profession. Professional ethos is a construct that results from combining elements such as the values, guidelines, and codes of professional ethics; a system of motivations that bind the subject to their professional practice; and forms of self-regulation of behaviour that maintain coherence between ethics and a professional morality to which one aspires [17–19]. This professional ethos, as a construct, is linked to a process that is always linked to the construction of the individual's own identity.

The relationship between the professional ethos of the teacher and the teaching practice is close and bidirectional. On the one hand, the professional ethos of the teacher influences the teaching practice, orienting it towards certain objectives and strategies. On the other hand, the teaching practice also contributes to the construction of the professional ethos of the teacher, reinforcing or modifying their values, principles, and beliefs [20]. In this article, the concept of "ethos" over "professional identity" is preferred because university practicum experiences are seen as a space for individual and social construction, generating observations, participation, and reflections with an ethical commitment to improving the educational reality. Additionally, according to Wanjiru [21], a professional ethos is formed by combining natural or vocational aptitudes with formal and personal requirements to enhance one's educational and professional performance. In this regard, Rojas [22] explains that ethical development occurs, allowing educators to weigh up the implications of their actions based on their commitment to themselves and society as a whole. Hence, the professional ethos encompasses individual factors such as commitment and responsibility, alongside social factors such as innovation and collaboration. In terms of social factors, Oser and Biedermann [23] point out the importance of understanding models of excellence, tradition, or normativity to generate virtues and moral dispositions. Similarly, Tirri [24] explains how routines, habits, practices, and challenges are essential in shaping the professional ethos.

This commitment is particularly relevant in challenging contexts, such as the one chosen for this study, which examines the effects of the pandemic on the development of professional ethos. In such contexts, certain expectations may be mediated or disrupted, and values related to commitment and responsibility in professional action could be altered. It is also important to consider, as argued by Beauchamp and Thomas [25], that there is no comprehensive framework for understanding teacher identity, as multiple factors contribute to its formation. Furthermore, some authors have debated the interaction between ethos and identity. For instance, Sutton contends that one's personal identity

generates one or several professional ethe, while Peloquin [26] argues that ethos guides the construction of professional identity by setting the course for a profession in times of change, emphasising the value of professional actions in terms of improvement. This ethos, as a space for improvement, responsibility, and integrity, is also influenced by the practicum students' tutors. They guide students towards ethical and responsible actions, serving as role models who shape both individual and collaborative behaviours.

Associated with the concept of professional ethos, we can find the concept of resilience, which can be understood as the ability to cope with stress, improve or maintain self-esteem, and learn from experience in situations that may, at first, be adverse. Levine [27] explored how resilience is the ability to utilise inner strengths and outer resources. This means being able to not only cope with work stress but also to manage and adapt to changes that may occur in professional activity as well as to overcome obstacles in the face of adversity [28]. The resilience and professional ethos concepts are connected. In fact, crises in the process of constructing professional identity are commonplace [16,19]. In the context of the pandemic, the systems of values and ethical codes, as well as the system of motivations of future teachers, were put to the test. That is why in this research work, we wanted to look deeper into the mechanisms that made it possible to maintain and/or reinforce all of these systems.

This comprehensive framework of variables shaping professional action leads us to the study of professional ethos and resilience by analysing the experiences of practicum students. Placing the study of this ethos within the context of a pandemic allows us to understand whether limitations in the development of educational activities may have impacted the initial development of their professional ethos.

### 1.1. The Pandemic as a Scenario for the Initial Development of the Professional Ethos of Teaching Students in Spain

The onset of the pandemic led to a lockdown that compelled both teachers and education students in Spain to adapt to new ways of teaching, such as online education. This required much greater flexibility and adaptability to meet students' needs. During the lockdown, educators had to innovate and adapt to virtual teaching environments, often resulting in increased stress and anxiety among teaching staff [29]. In subsequent stages, as physical distancing restrictions were imposed to prevent infections, the teaching–learning processes in schools were also affected. Educators had to demonstrate increased flexibility and develop new skills to devise novel teaching methods or respond to the socio-emotional and physical health needs of their students [30,31].

In Spain, several decree-laws and regional measures imposed limitations in both primary schools and preschools, where teaching students were required to complete their practicums. At the national level, during the 2020–2021 academic year, recommendations were issued for education centres in Spain, focusing on the following aspects: prioritising the use of outdoor spaces; organising students into stable groups beyond the first cycle of Primary Education; improving communication with families by phone, email, messaging, or regular mail, facilitating administrative procedures electronically; recommending the promotion of active transportation (walking or cycling) to school; and taking action to prevent stigma or discrimination related to COVID-19, with special attention on cases of greater emotional and social vulnerability that may have arisen as a result of the pandemic. The primary goal of these measures was always to ensure the health and safety of both staff and students in the face of the COVID-19 coronavirus. To achieve this, preventive actions focused on two main points:

- Preventing infections.
- The early detection and isolation of potential cases that could occur within the centres.

Based on these general criteria, the education departments of various autonomous communities established specific protocols that required schools to reorganise their spaces, both academically and recreationally, suppressing all extracurricular or complementary activities. Moreover, all pathways were clearly marked to avoid any form of contact or proximity. It is worth noting that in Spain, educational competencies come under the

jurisdiction of autonomous communities, which oversee the operation of non-university educational institutions.

The main organisational measures adopted by these regional institutions were as follows: In Early Childhood Education (ages 3–6), students were organised into stable coexistence groups, adhering to the ratios established in applicable regulations, with the guidance of the classroom teacher. In Primary Education, for 1st to 4th grades, student organisation was based on stable coexistence groups. For 5th and 6th grades, organisation was also carried out based on these stable coexistence groups, maintaining a distance of 1.5 m, with the possibility of flexibility down to 1.2 m.

Considering that the practicum for students in the Teacher Education programme at the University of Salamanca primarily took place in the autonomous community of Castilla y León, here is a summary of the most significant interventions related to COVID-19 during the 2020–2021 and 2021–2022 academic years in non-university educational institutions located in the region:

- Two main types of classrooms were established: stable coexistence groups (composed of students from the first and second cycles of Early Childhood Education, as well as those in the 1st grade of Primary Education) and all other groups.
- The mandatory use of masks for all (covering the nose and mouth) was required, except for Early Childhood Education and the 1st grade of Primary Education, where it was voluntary.
- Hydroalcoholic gel dispensers and $CO_2$ monitors were placed in classrooms, maintaining a safety distance of 1.5 m (both during movements and in the arrangement of chairs and desks in various spaces), and face-to-face seating within the educational community was not permitted. Before any educational material was used, it was mandatory to disinfect hands with the gel. Additionally, classrooms were required to be ventilated in case of excessively high $CO_2$ levels and between different class time slots.
- Multiple entrances to the schools were made available, with pathways and corridors being marked. In addition, separate schedules and spaces were designated for break time for each level and education cycle.
- The use of shared materials in any area was not allowed, and it was essential to rely on individual tools and personal work resources.

In the event of a potential infection case, the intervention protocol required reporting it to the provincial COVID-19 committee to conduct relevant contagion verification tests (testing all members of stable groups and only the affected individuals and their close contacts in other groups). If a positive case was confirmed, the entire stable coexistence group or the specific positive cases in other groups were required to quarantine at home. Initially, this home quarantine period was fifteen days, but was later reduced to ten. In the 2021–2022 academic year, tests were carried out by those directly involved or interested, without the need for testing the entire classroom, and quarantine decisions were made by health authorities at health centres. During the 2021–2022 academic year, organisational protocols for the distribution of spaces and people were maintained, along with the basic principles of prevention, hygiene, and health promotion against COVID-19 in educational institutions. These principles can be summarised as follows:

1. Limiting contacts: maintaining physical distance and organisation based on stable coexistence groups (SCG).
2. Personal prevention measures: hand hygiene, mandatory mask use from the age of 6, and vaccination of the education community.
3. Cleaning and disinfection combined with continuous cross-ventilation.
4. Case management following established protocols, ensuring coordination between Health and Education authorities.

As of May 2022, the mandatory use of masks in common spaces was lifted, and the gradual resumption of some basic complementary activities began.

*1.2. Functions and Tasks of Practicum Students in Teaching Degree Programmes in Spain*

In Spain, the Bachelor's Degree in Early Childhood Education and the Bachelor's Degree in Primary Education are linked to the professional activity for which students are prepared, in accordance with the general principles established by Article 3.5 of the Royal Decree 1393/2007 of 29 October. These programmes must demonstrate the competencies outlined in the annexes of Orders ECI/3854/2007 and ECI/3857/2007, which also contain the requirements, content, and tasks to be developed during their practicum processes. In teaching degree programmes, the practicum is identified as a set of activities designed for students to acquire the professional responsibility required to directly engage with education institutions, according to the following functions:

1. Acquire practical knowledge of the classroom and its management.
2. Understand and apply interaction and communication processes in the classroom, mastering the social skills necessary to foster a classroom climate that is conducive to learning and coexistence.
3. Monitor and track the education process, particularly the teaching–learning process, through the mastery of certain required techniques and strategies.
4. Relate theory and practice with the reality of the classroom and school.
5. Engage in teaching activities and learn how to perform, act, and reflect based on practice.
6. Participate in proposals for improvement in various areas that may be established within a school.
7. Regulate interaction and communication processes in groups of students aged 3–6 and 6–12 years.
8. Understand methods of collaboration with the different sectors of the education community and the broader social environment.

*1.3. Purpose and Objectives of the Study*

This study aims to analyse how restrictions in the context of the COVID-19 pandemic and the exceptional periods that mandated preventive measures in educational institutions affected teacher practicums throughout the 2020–2021 and 2021–2022 academic years. The objective is to gain an insight from the student's perspective into the limitations and opportunities that emerged during their practicum experiences in this extraordinary context to see the effects on the construction of their professional ethos.

To achieve this purpose, the study is divided into the following objectives:

1. To analyse students' **expectations** prior to the start of their practicum and determine if there were differences in the perception of limitations and opportunities in the development of their functions depending on social variables such as gender and academic year.
2. To differentiate students' **perceptions** in their discourse regarding functions that were enhanced and limited while establishing gender and academic year differences.
3. To comprehend the **subjective effects** of external pandemic-imposed limitations on the practicum.

## 2. Materials and Methods

*2.1. Population and Selection Method*

This study focused on a population of students pursuing Bachelor's Degrees in Early Childhood Education, Primary Education, and Double Degrees, from three campuses of the University of Salamanca, who carried out their practicums in schools located throughout Spain. The sample was selected for convenience over two academic years (2020–2021 and 2021–2022) from participants in Practicum I and II, in their third and fourth years, respectively. A total of 725 students voluntarily and with their consent participated in the study, comprising 584 women (80.55%) and 141 men (19.45%), with a mean age of $x_{age}$ = 22.15 years and a standard deviation of 2.82. Throughout the data collection and anal-

ysis process, ethical standards mandated by the University's regulatory committees were adhered to, as well as compliance with both Spanish and European Data Protection Laws.

### 2.2. Instrument

The instrument employed is a questionnaire consisting of ad hoc qualitative nominal and ordinal questions, as well as open-ended questions. In our detailed analyses, we primarily focus on the responses to the open-ended questions, which inquired: "How did you expect the pandemic and restrictions to affect your Practicum? Have those expectations been met? Why?", "What competencies do you believe the completion of the Practicum in the COVID-19 context has provided you, aside from those previously mentioned?", and "Which competencies have been most adversely affected by having to complete the Practicum during COVID-19 times?" The academic year (2020–2021 and 2021–2022) and gender (male and female) variables from the questionnaire are also incorporated.

### 2.3. Analysis

The analysis focuses on the responses to open-ended questions posed after the completion of practicum periods by third-year (Practicum I) and fourth-year (Practicum II) students, as per the aforementioned questionnaire. To achieve this, a Statistical Analysis of Textual Data (SATD) was employed using the MANOVA Biplot and Reinert's Alceste methods [32–34]. The SATD conducted begins with the creation of a lexical table [35,36] with the assistance of the IRaMuTeQ programme version 0.7. This table constitutes an $X_{nxp}$ matrix with n participants represented in the rows of the lexical table using a unique identifier (ID) and p words in the columns. This matrix was simplified using the lemmatisation protocol [35,36], which involves converting verbs to their infinitive form and nouns and adjectives to their masculine singular form. Initially, using the same software, the calculation of word categories ("classes" in Alceste terminology) was requested, along with their graphical representation, in the form of Multiple Correspondence Factorial Analysis [37,38] and Hierarchical Cluster Analysis dendrograms [32]. Subsequently, utilising the lexical table generated by IRaMuTeQ, the lexical data matrix was subjected to the Characterisation Value proposed by Caballero et al. [39–42], which allows for reweighting of the frequencies in the lexical table by readjusting their weights based on their qualitative characterisation. The resulting matrix was then processed using a two-way MANOVA Biplot or Canonical Biplot [43,44], enabling the presentation of differences in the discourses of various groups or collectives. The formation of these groups was generated based on the categorical variables "academic year" and "gender".

## 3. Results

### 3.1. Optimism in Times of Hardship

In the open-ended responses to questions about the expectations of male and female students regarding the practicum, "How did you expect the pandemic and restrictions to affect your Practicum? Have those expectations been met? Why?", the SATD using the Reinert method reveals the presence of six major thematic blocks or aspects as identified by the students (see Figure 1).

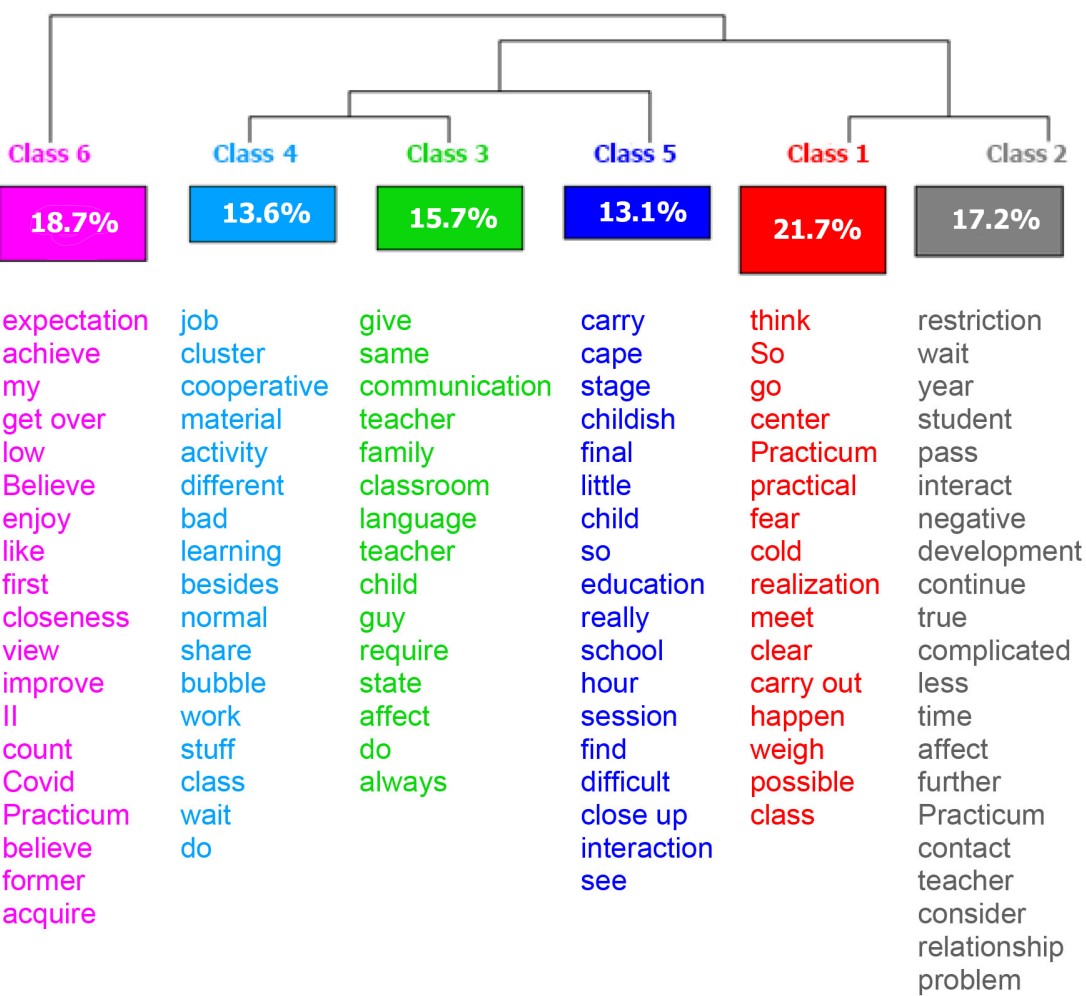

**Figure 1.** Classification of students' expectations during COVID-19 using the Reinert method.

This analysis reveals an initial separation of discourses into two perspectives: one with a negative aspect influenced by the fears and difficulties of the pandemic, and another with a more positive and optimistic approach to facing the practicum. These discourses are further divided into different themes (classes). Class 1 (red) encompasses discourses in which students use words to refer to their fears when it comes to conducting the practicum in the midst of the COVID-19 pandemic. On the other hand, Class 2 (grey) illustrates the expectations of a complex situation dominated by pandemic restrictions and their negative effects on both student development and the practicum. In short, these two classes capture the discomfort and unease caused by the global health situation. On the other hand, from a more optimistic perspective, we have Classes 3 (green), 4 (light blue), and 5 (dark blue), which showcase communicative and interactive teaching efforts despite the difficulties. Class 3 emphasises teacher communication and maintaining communicative quality with students in Early Childhood and Primary Education, as well as with their families. Meanwhile, Class 4 underscores cooperative and shared work that allows for working as normally as possible. Class 5, however, highlights the importance of and the need for establishing a connection and maintaining interaction in these stages of education. Additionally, in this discourse, the words "stage" and "childish" stand out, suggesting a greater emphasis on this stage of education. Finally, Class 6 presents a generic and optimistic discourse in which it is stated that their experiences allowed them to exceed their expectations during the practicum.

On the other hand, the SATD analysis using the MANOVA Biplot allows us to differentiate each group (men or women in each practicum period) by identifying the most characteristic discourse (set of words) for each one. The resulting graph displays these

characteristic words through vectors, indicating their significance for each group. Represented as small points on the plane with a circle delimiting the 95% confidence limits, each group moves closer to or farther away from the others depending on their level of (dis)similarity. The vectors, in turn, indicate which words cause these (dis)similarities. Figure 2 clearly distinguishes between the discourses offered by men and women. Similarly, it is apparent that men's discourse changed depending on the year of the practicum, while women maintained a consistent discourse across both academic years. As seen in the figure, the groups of women for each course closely cluster around the coordinate centre, whereas those of the men move towards the first quadrant in green (expectations of men in the 2020–2021 academic year) and the second quadrant in yellow (expectations of men in the 2021–2022 academic year).

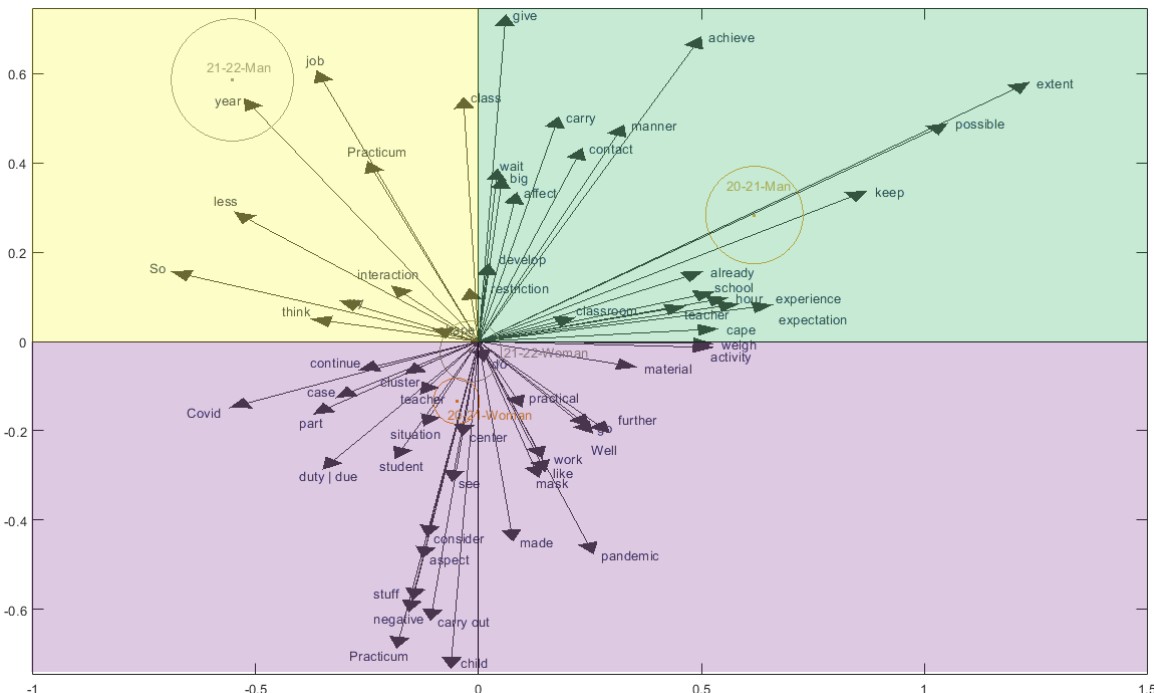

**Figure 2.** SATD through MANOVA Biplot of students' expectations during COVID-19.

Therefore, based on the academic year and gender variables, three distinct discourses characterise the different groups. Women, following a more homogeneous discourse over time, differentiate themselves from men by maintaining a discourse in which negative aspects of the restrictions prevail, such as working with a mask during the pandemic or monitoring cases of infections.

*It has affected interactions with the students due to the use of masks, interactions with other teachers, lesson planning and organisation, schedules, and many other aspects of daily life in the classroom. I expected all of these aspects.* (Female, 22 years old, 5th year Double Degree in Early Childhood and Primary Education, specialising in Religion)

*I knew it would have an impact since many types of activities cannot be carried out. In my case, as I was doing my practicum in Special Education, wearing masks had a negative impact on teaching language because the children couldn't see how we position our mouths when producing different phonemes. Also, I wanted to see how cooperative group work is, and since we had to sit individually, I could not work with this methodology.* (Female, 22 years old, 4th year Bachelor's Degree in Primary Education, specialising in Special Education)

Meanwhile, the men who responded in the 2020–2021 academic year characterise their discourse by a concern for the measures imposed in the classroom that limited contact and the use of materials.

*I expected a different experience due to the many restrictive measures in place. These expectations have been met since cooperative activities that involved grouping students couldn't be carried*

*out.* (Male, 21 years old, 4th year Bachelor's Degree in Primary Education, specialising in Generalist Education)

In 2021–2022, their concern derived from the possibility of limiting interaction and work in the classroom.

*Students' work in class has been different than before COVID-19.* (Male, 20 years old, 3rd year Bachelor's Degree in Primary Education, specialising in Physical Education)

### 3.2. Adaptation as a Competence in Response to the Pandemic

When asked, "What competence/s do you think the completion of the Practicum in the COVID-19 context has provided you with, apart from those previously mentioned?", students once again showed gender and course-related differences (see Figure 3). On one hand, during the 2020–2021 academic year, male students emphasise purely didactic aspects, such as competencies for adapting to the various difficulties and situations that arose during the pandemic. In this regard, communicative and methodological aspects are highlighted as essential parts of this discourse.

*Programming the didactic unit with a particularly individual methodology.* (Male, 21 years old, 4th year Bachelor's Degree in Primary Education, specialising in Generalist Teaching)

*Highlighting what motivates students to learn despite the present difficulties.* (Male, 22 years old, 4th year Primary Teacher, specialising in Physical Education)

On the other hand, women also emphasise the ability to adapt to changing situations. However, they lack the didactic aspect that characterises the male group for the same academic year. Women, in contrast, highlight adaptation oriented towards group management, conflict resolution, and hygiene.

*Conflict resolution and managing my work with a large group of students.* (Female, 23 years old, 3rd year in Early Childhood Education, specialising in Special Education)

*The ability to adapt to changes, managing insecurity, planning activities with limitations…* (Female, 20 years old, 3rd year Bachelor's Degree in Primary Education, specialising in Generalist Teaching)

*It has served to work more consciously on hygiene habits and respect for others.* (Female, 22 years old, 5th year Double Degree in Early Childhood Education and Primary Education, specialising in Generalist Teaching)

By the 2021–2022 academic year, their discourse evolved towards the implementation of safety measures for the classroom and their students. Meanwhile, male students progressed to the next academic year acquiring a less differentiated and heterogeneous discourse, sharing characteristics with all the previous ones.

*I have been able to observe the impact of the situation, as well as safety measures among the students and their development.* (Female, 22 years old, 3rd year Bachelor's Degree in Early Childhood Education, specialising in Special Education)

*Programming activities for children aimed at greater safety and space management to maintain the required distance and necessary measures, as well as recognising the importance of instilling hygiene habits in children to prevent diseases, such as COVID-19 at this time.* (Female, 20 years old, 3rd year Bachelor's Degree in Primary Education, specialising in Special Education)

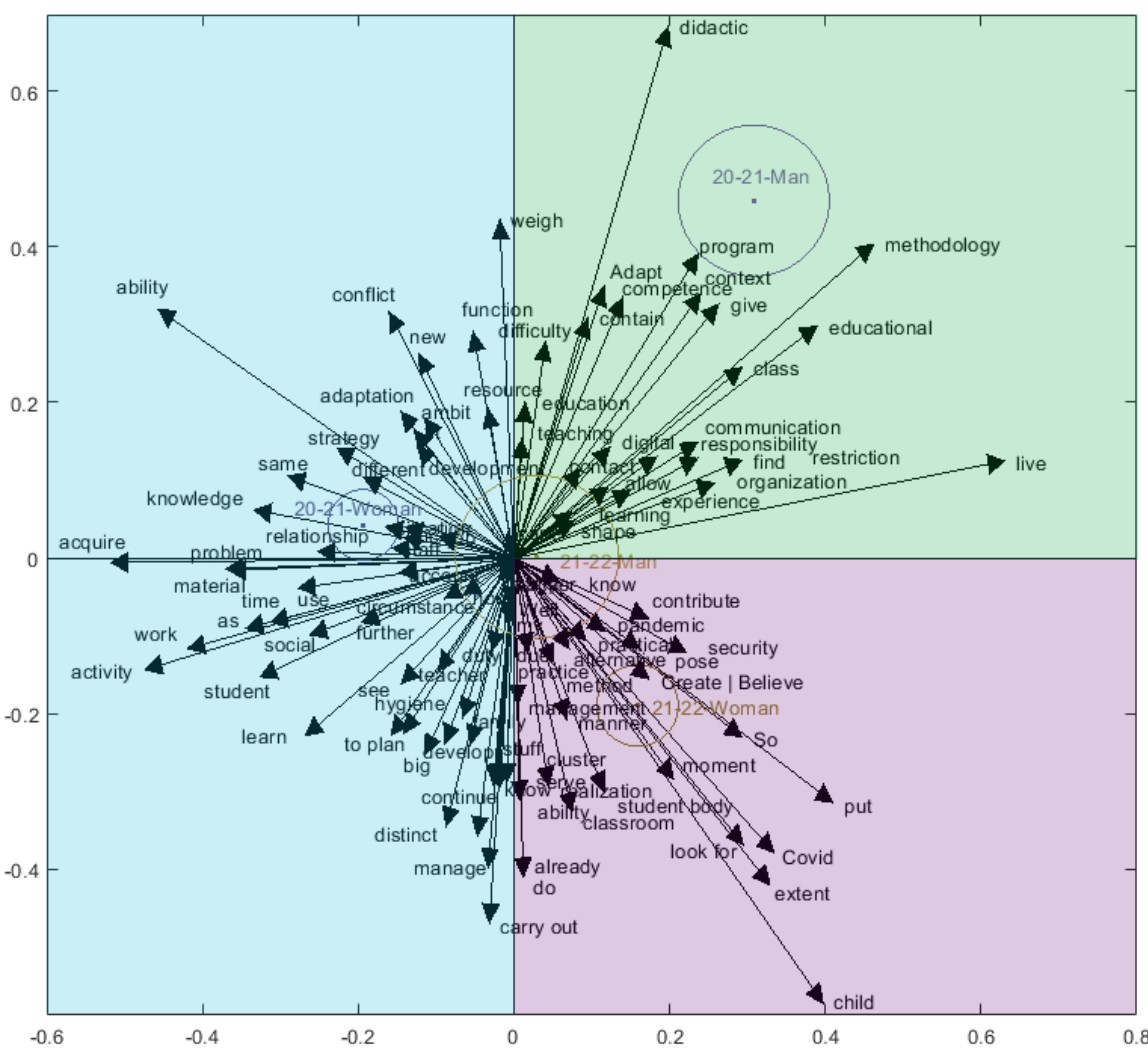

**Figure 3.** SATD through MANOVA Biplot of acquired competencies.

### 3.3. Interaction as the Major Affected Aspect

When asked, "Which competence/s have been most affected by having to carry out the Practicum during COVID-19 times?", the SATD through MANOVA Biplot reveals different discourses in which, in one way or another, interpersonal relationships are at the centre of the competencies that were most affected during the pandemic. In this way, we can see how women, during the 2020–2021 academic year, express concerns about group and collaborative aspects in which interaction with students and families was severely affected (see the second quadrant of Figure 4).

*There hasn't been as much interaction as there could have been with the students, especially with the families. Additionally, there are certain activities that couldn't be done, like festivals or more practical activities.* (Female, 20 years old, 3rd year Bachelor's Degree in Primary Education, German)

*As for the affected competence, I would say everything related to teamwork, meaning cooperative work. On the other hand, we also couldn't move to different classrooms or observe different groups, and that's one of the most relevant aspects of the practicum.* (Female, 24 years old, 4th year Bachelor's Degree in Early Childhood Education, specialising in Generalist Teaching)

In the 2021–2022 academic year, women shifted their discourse towards the more communicative aspects of interaction, insisting on linguistic and teamwork competencies (see the first and fourth quadrants).

*Competences in which cooperation was needed, from group activities to more personal interaction with students.* (Female, 21 years old, 4th year Bachelor's Degree in Primary Education, General Education)

*The communicative competence is the most affected, as components such as tone of voice or oral fluency have been significantly different due to the use of masks indoors, affecting the interaction between the teachers and students.* (Female, 22 years old, 4th year in Primary Education, specialising in Audition and Language)

Lastly, while men maintained an intermediate discourse relative to these two perspectives during the first analysed year, in the second year, they expressed concerns about the barriers posed by surgical masks to the teacher's approach and social interaction.

*Interacting with students has been challenging due to the use of masks, as it requires raising one's voice significantly, and sometimes, it's still difficult to understand the students.* (Male, 22 years old, 3rd year Bachelor's Degree in Primary Education, Physical Education)

*Interaction in the classroom, particularly regarding the ability to implement methodologies that involve grouping students or using complementary materials, has been affected.* (Male, 23 years old, 4th year Bachelor's Degree in Primary Education, General Education)

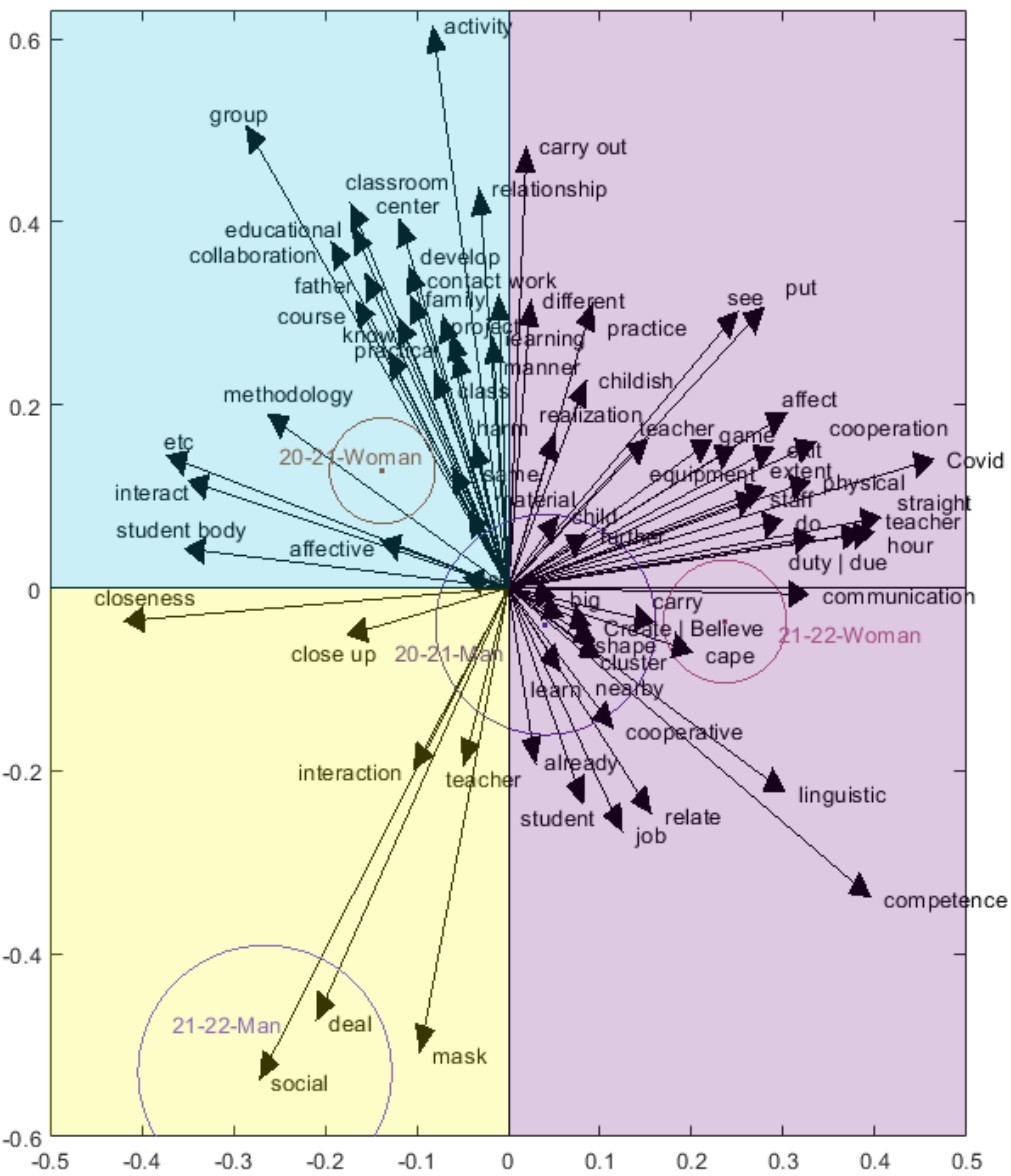

**Figure 4.** SATD through MANOVA Biplot of affected competencies.

## 4. Discussion and Conclusions

This descriptive study is part of the broader discussion on teachers' professional development through reflective practice. Therefore, the treatment or development of the selected cases is argued by examining moments of reflection and stimulating the recollection of students' training to become teachers. This reflective analysis is prompted by the questions in the questionnaire, as detailed in the intervention methodology. As such, it represents a reflection on their perception of the practicum as a space that shapes their professional ethos in a moment of unprecedented challenges posed by the pandemic, which is presented as an educational and professionalising reality [6,7].

In line with what was indicated in the introduction, this practical and reflective dynamic has influenced—by facilitating or hindering—the process of self-construction of the students' professional ethos. First, the experiences and routines developed during their internships and the external constraints imposed by the pandemic determined their professional performance and ethos. This entire reflective context was verified through the discourses of the future teachers. This study found that of the three variables proposed by Canton [2] (expectations, experiences, and routines), routines were the most determinant factor impacted by the pandemic constraints. This means that for future teachers, not only was the pandemic a health crisis, but it was also a crisis that directly affected the process of their professional identity construction [17,18] and their resilience to such adversities. According to the data we found, expectations were, in general, positive and oriented towards an optimistic attitude despite the fear. This shows, therefore, that future teachers were able to respond to the situation with a high degree of resilience, despite the fact that women tended to show more negative expectations than men.

As has been indicated, the historical context within which the construction of habitus and professional ethos takes place is of vital importance when it comes to understanding these concepts and being able to analyse them [9,15]. The very special conditions offered by the pandemic invited us to take an interest in the effects it could have on future teachers. Based on the same logic, the timing of the questionnaire over two academic years, conditioned by the restrictions of COVID-19, had a direct impact on the responses related to communication between the student teachers, their students, and the professional tutors where the internships were carried out. This was especially noticeable in the context of educational planning and organization. Here another ethos formation variable is reflected, related to collaboration and models of excellence, as described by Oser and Biedermann [23].

Regarding commitment and responsibility at an individual level, the high degree of intrinsic motivation that comes with conducting practicums fostered a more positive view of the identified learning. This pertains to the individual perceptions expressed in the responses (on a personal basis) of the surveyed Teacher Training students regarding the association of resilience in their identity processes with the profession, concerning their professional ethos. Regardless of the collective results, there is a clear, implicit optimism towards adaptability as a competence in dealing with the pandemic. As Luna [13] concludes, "it is possible to modify certain aspects of the professional teaching ethos, rebuilding it towards a set of values, assumptions, motivations [...]. The process to achieve this is disruptive to the status quo and involves communication, awareness-raising, among others". This optimism reflects a perception of educational opportunity amidst the challenges posed by the pandemic, as supported by the statistical data obtained through the MANOVA Biplot instrument. These positive attitudes from the participants are also related to the optimistic and hopeful attitudes required for transformative actions because, as described by Wamsler et al. [28], the attitudes empower people in agency when the ability to see and understand deeper patterns leads to changes in our own role to transform the challenge into learning opportunities.

However, it is noteworthy that male students tend to lean more towards issues related to their didactic capacities as educators, while female students are particularly inclined towards matters linked to group management, highlighting limitations that hindered them

from fulfilling their social responsibility. This suggests that, within the social space in which the teacher's professional ethos is constructed, there are gendered socializations [14,15,45] that affect the set of values that constitute each of these ethe. In any case, they mostly point out that interaction was greatly affected. However, certain details suggest that students acquired valuable skills through their experience, learned new methodologies, and invented new methods for the continuous monitoring of specific students, especially those with learning difficulties. Thus, there is a general concern among students to efficiently carry out tasks that enable systematic observation of the most significant evidence resulting from their participation in Early Childhood and Primary Education classrooms during their practicums, all while adhering to objective criteria. Therefore, despite the inherent difficulties of the circumstances in the 2020–2021 and 2021–2022 academic years, a resilient attitude allowed for the implementation of creative solutions and innovations, resulting in gains for the initial formation of their professional ethos.

We can conclude that the pandemic has had a significant impact on the professional ethos of Teacher Training students, requiring them to show greater adaptability, flexibility, empathy, emotional care, innovation, and creativity. These changes in the value system may remain relevant even after the pandemic as they constitute an enduring system of dispositions embedded in the ethos of the future teacher [8–10,13]. This contributes to the improvement of education in the future from a resilient and transformative perspective. The results of the questionnaires clearly indicate two key aspects. On the one hand, they reflect the initial discomfort resulting from the difficulties of implementing COVID-19 health measures in schools, which translates into a reduction in face-to-face teaching activities with students. However, this initial discomfort has been overcome, affirming that the teaching–learning processes have provided them with strategies and tools that would not have emerged in a normal context. Resilience has enhanced the formation of their ethos as educators who face challenges with commitment and a sense of responsibility for improvement. Levine [27] explored how resilience is the ability to utilise inner strengths and outer resources, confirmed in our study since participants combined their individual abilities to confront social restrictions.

To conclude, it is worth emphasising that this analysis of specific cases paves the way for reflective intervention in school practicum processes within Teacher Training programmes. These findings can serve as a foundation for future initiatives that can be extrapolated to other regions and universities across the country with similar curriculum structures. However, it is essential to note certain limitations. Firstly, there is a limitation concerning the interpretation of results related to gender, given that Teacher Training programmes exhibit a strong female presence, although this is a consistent trend in studies on teacher training programmes [46].

Another limitation is the relatively small number of Teacher Training students analysed who completed their practicum in rural settings. In these environments, the specific measures related to COVID-19's spatial organisation, during the 2020–2021 and 2021–2022 academic years, were implemented in a less drastic manner due to the peculiarities of classrooms in many villages in Castilla y León (with a limited number of students). This allowed hygiene and distancing issues to be less acute compared to urban school settings (where class sizes are much more constrained). In other words, the perspectives on interventions may differ, taking into account the diversity of spaces and circumstances within the broad rural spectrum in some areas of Spain.

**Author Contributions:** Conceptualization, M.J.H.-S.; methodology, D.C.-J.; formal analysis, D.C.-J.; investigation, M.J.H.-S., D.C.-J., and A.S.-M.; resources, A.S.-M. and L.H.-P.; data curation, A.S.-M. and L.H.-P.; writing—original draft preparation, M.J.H.-S., D.C.-J., A.S.-M., and L.H.-P.; writing—review and editing, M.J.H.-S., D.C.-J., A.S.-M., and L.H.-P.; supervision, M.J.H.-S.; project administration, M.J.H.-S.; funding acquisition, M.J.H.-S. All authors have read and agreed to the published version of the manuscript.

**Funding:** This research received no external funding.

**Institutional Review Board Statement:** In this study, ethical review and approval were waived due to the voluntary participation of the subjects, the anonymity of the responses collected, and the consent provided by the study participants.

**Informed Consent Statement:** Informed consent was obtained from all subjects involved in the study.

**Data Availability Statement:** Data are contained within the article.

**Acknowledgments:** This study was developed under the framework of the project: Banco de casos prácticos basados en el Modelo de Competencias Profesionales del Docente de Castilla y León para la mejora de la formación del Prácticum de los Grados de Maestro (ID2021/153). PI: María Jose Hernández-Serrano. Supported by: Dirección General de Innovación y Formación del Profesorado, de la Junta Castilla y León (Spain).

**Conflicts of Interest:** The authors declare no conflicts of interest.

## Abbreviations

The following abbreviations are used in this manuscript:

| | |
|---|---|
| SATD | Statistical Analysis of Textual Data |
| COVID-19 | Coronavirus disease |
| MANOVA Biplot | Multivariate Analysis of Variance |
| IRaMuTeq | Interface de R pour les Analyses Multidimensionnelles de Textes et de Questionnaires |
| SCG | Stable coexistence groups |

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
