# Peer review of "Professional Ethos and Resilience in the Identity of Practicum Students during the Pandemic Context"

_education, doi:10.3390/educsci13121207_

Round 1

Reviewer 1 Report

Comments and Suggestions for Authors Thank you for the interesting research angle and methodology!

The abstract and introduction should clearly state the aim and objectives of the research, and the research question/hypothesis.

Currently, research categories are revealed only in  the section 1.3.

No research questions or hypothesis have been formulated.

The reference is missing in line 39 of the text.

The part of scientific literature, research analysis should be strengthened. The literature analysis should support the understanding of Professional ethos and Resilience in this research.

Findings and conclusions would be more conceptualized in terms of benefits in the post-pandemic era.

All items in the reference list should be presented in the same style; please pay special attention to 10th item in the reference list.

Author Response

Dear reviewer, thank you for taking the time to review our work. We have read your comments and have made the following changes to the article:

  1. Thank you for the interesting research angle and methodology!

Thank you too for your time reviewing our work.

  1. The abstract and introduction should clearly state the aim and objectives of the research, and the research question/hypothesis. Currently, research categories are revealed only in the section 1.3. No research questions or hypothesis have been formulated

We have improved the redaction of the research question, aim and objectives of the research.

  1. The reference is missing in line 39 of the text.

You can find now in line 39 of the text the reference number 4 Asensio-Muñoz, I.; Ruiz-Miguel, C. Medida y evaluación de las creencias sobre la profesión de
los maestros en formación. Revista Electrónica Interuniversitaria de Formación del Profesorado 2017,20, 79. https://doi.org/10.6018/reifop.20.3.265231.

There was an error that has been modified.

  1. The part of scientific literature, research analysis should be strengthened. The literature analysis should support the understanding of Professional ethos and Resilience in this research.

We have reinforced the literature on professional ethos and resilience in the introduction and the discussion of findings.

  1. Findings and conclusions would be more conceptualized in terms of benefits in the post-pandemic era.

Thank you for your comments. We have improved the redaction of these sections by combining the literature added to our theoretical background and the findings of our study.

  1. All items in the reference list should be presented in the same style; please pay special attention to 10th item in the reference list.

We have rectified the error.

Reviewer 2 Report

Comments and Suggestions for Authors

Thank you very much for raising a topical issue in teacher education – the formation of students’ professional ethos during the two academic years affected by the pandemic. The assessment discourse of the practicum offered in university degree programmes has been analysed. An essential conclusion has been drawn that “the practicum experiences during the pandemic-affected academic years facilitated the development of students’ professional ethos, emphasizing resilient attitudes and solutions”. It is important for pre-service teachers to have the opportunity to develop their professional skills and competence concerning the formation of a teacher`s identity.

The introduction part reveals the theoretical grounding of the formation of professional ethos by emphasizing the significance of internship periods within teacher education programmes in the context of the pandemic. A very detailed description is given regarding the conditions that were present during the pandemic period. The author(s) aim to understand the restrictions and their impact on the formation of professional ethos.

However, it is not clear why the title of the article refers to the “post-pandemic era”, but the study was carried out in 2020-21 and 2021- 22 academic years, that is, during the pandemic. Also, the purpose and objectives of the study are not clearly formulated, thus, causing confusion about what the purpose and objectives really are in the context of the topic of the article, for example: 

“study aims to conduct an evolutionary analysis of how restrictions affected teacher practicums…”, the objective is to gain insight from the student’s perspective into the limitations and opportunities that emerged during their practicum experiences…”, “the purpose has been divided into several study objectives...”. Moreover, the objectives that are put forward in the text, include an analysis establishing differences by students’ gender and academic year – however, these aspects are not addressed in the analysis of the theory. Also, it is not clear from the wording, how the defined purpose and objectives are related to professional ethos and identity.

I recommend giving a clearer wording/ formulation of the purpose and objectives as well as supplementing the theoretical analysis with the aspects of professional ethos that are related to students' gender and academic year.

The methodological part is clearly stated and the results are described sequentially and transparently. The discussion and conclusions describe the main findings and limitations of the study. However, this part includes only general assumptions about how the research results could be used in the future - in the post-pandemic era. For now, it is concluded that “the pandemic has had a significant impact on the professional ethos”, “the analysis of specific cases paves the way for reflective intervention in school practicum processes”, etc.

The research would only benefit if the conclusions could point more specifically exactly how the results obtained during the research could improve the teacher education process and the formation of professional ethos nowadays, in the post-pandemic period.  

Author Response

Dear reviewer, thank you for taking the time to review our work. We have read your comments and have made the following changes to the article:

  1. Thank you very much for raising a topical issue in teacher education – the formation of students’ professional ethos during the two academic years affected by the pandemic. The assessment discourse of the practicum offered in university degree programmes has been analysed. An essential conclusion has been drawn that “the practicum experiences during the pandemic-affected academic years facilitated the development of students’ professional ethos, emphasizing resilient attitudes and solutions”. It is important for pre-service teachers to have the opportunity to develop their professional skills and competence concerning the formation of a teacher`s identity.
    1. Thank you for your comments and your time to review our article.
  2. The introduction part reveals the theoretical grounding of the formation of professional ethos by emphasizing the significance of internship periods within teacher education programmes in the context of the pandemic. A very detailed description is given regarding the conditions that were present during the pandemic period. The author(s) aim to understand the restrictions and their impact on the formation of professional ethos.
    1. We have reinforced the literature on professional ethos and resilience in the introduction.
  3. However, it is not clear why the title of the article refers to the “post-pandemic era”, but the study was carried out in 2020-21 and 2021- 22 academic years, that is, during the pandemic. Also, the purpose and objectives of the study are not clearly formulated, thus, causing confusion about what the purpose and objectives really are in the context of the topic of the article, for example: “study aims to conduct an evolutionary analysis of how restrictions affected teacher practicums…”, the objective is to gain insight from the student’s perspective into the limitations and opportunities that emerged during their practicum experiences…”, “the purpose has been divided into several study objectives...”. Moreover, the objectives that are put forward in the text, include an analysis establishing differences by students’ gender and academic year – however, these aspects are not addressed in the analysis of the theory. Also, it is not clear from the wording, how the defined purpose and objectives are related to professional ethos and identity. I recommend giving a clearer wording/ formulation of the purpose and objectives as well as supplementing the theoretical analysis with the aspects of professional ethos that are related to students' gender and academic year.
    1. We have improved the redaction of the research question, aim and objectives of the research.
  4. The methodological part is clearly stated and the results are described sequentially and transparently. The discussion and conclusions describe the main findings and limitations of the study. However, this part includes only general assumptions about how the research results could be used in the future - in the post-pandemic era. For now, it is concluded that “the pandemic has had a significant impact on the professional ethos”, “the analysis of specific cases paves the way for reflective intervention in school practicum processes”, etc.
    1. We have improved the redaction of these sections by combining the literature added to our theoretical background and the findings of our study.
    2. We have reinforced the literature on professional ethos and resilience in the introduction and the discussion of findings.
  5. The research would only benefit if the conclusions could point more specifically exactly how the results obtained during the research could improve the teacher education process and the formation of professional ethos nowadays, in the post-pandemic period.  

In conclusion:

  1. We have improved the redaction of the research question, aim and objectives of the research.
  2. We have reinforced the literature on professional ethos and resilience in the introduction and the discussion of findings.
  3. We have improved the redaction of these sections by combining the literature added to our theoretical background and the findings of our study

Reviewer 3 Report

Comments and Suggestions for Authors

This study examined the impact of the COVID-19 pandemic on the development of professional ethos in teacher education programs, involving 725 students over two academic years. It presents that limitations on routines due to the pandemic significantly affected the students' professional ethos, emphasizing the importance of adaptability and resilience. Additionally, gender differences revealed varying expectations, with female students having more negative expectations and male students showing a positive evolution in their learning expectations over the years. This is a very well written paper. Congratulations!

Author Response

Dear reviewer, thank you for taking the time to review our work. We have read your comments and we are very pleased.  Thank you very much for your words!!! We have made changes to the article to improve its quality following the other reviewers' comments.